# Disease Manifestations in Mucopolysaccharidoses and Their Impact on Anaesthesia-Related Complications—A Retrospective Analysis of 99 Patients

**DOI:** 10.3390/jcm10163518

**Published:** 2021-08-10

**Authors:** Luise Sophie Ammer, Thorsten Dohrmann, Nicole Maria Muschol, Annika Lang, Sandra Rafaela Breyer, Ann-Kathrin Ozga, Martin Petzoldt

**Affiliations:** 1Department of Paediatrics, International Centre for Lysosomal Disorders (ICLD), University Medical Centre Hamburg-Eppendorf, 20246 Hamburg, Germany; muschol@uke.de (N.M.M.); annika.lang@stud.uke.uni-hamburg.de (A.L.); s.breyer@uke.de (S.R.B.); 2Department of Anaesthesiology, University Medical Centre Hamburg-Eppendorf, 20246 Hamburg, Germany; t.dohrmann@uke.de (T.D.); m.petzoldt@uke.de (M.P.); 3Department of Paediatric Orthopaedics, Children’s Hospital Altona, 22763 Hamburg, Germany; 4Department of Orthopaedics, University Medical Centre Hamburg-Eppendorf, 20246 Hamburg, Germany; 5Department of Medical Biometry and Epidemiology, University Medical Centre Hamburg-Eppendorf, 20246 Hamburg, Germany; a.ozga@uke.de

**Keywords:** mucopolysaccharidosis, MPS, disease manifestations, symptoms, morbidity, spine disease, anaesthesia, airway, perioperative complications, surgery

## Abstract

Patients with mucopolysaccharidoses (MPS) frequently require anaesthesia for diagnostic or surgical interventions and thereby experience high morbidity. This study aimed to develop a multivariable prediction model for anaesthesia-related complications in MPS. This two-centred study was performed by retrospective chart review of children and adults with MPS undergoing anaesthesia from 2002 until 2018. We retrieved the patients’ demographics, medical history, clinical manifestations, and indication by each anaesthesia. Multivariable mixed-effects logistic regression was calculated for a clinical model based on preoperative predictors preselected by lasso regression and another model based on disease subtypes only. Of the 484 anaesthesia cases in 99 patients, 22.7% experienced at least one adverse event. The clinical model resulted in a better forecast performance than the subtype-model (AICc 460.4 vs. 467.7). The most relevant predictors were hepatosplenomegaly (OR 3.10, CI 1.54–6.26), immobility (OR 3.80, CI 0.98–14.73), and planned major surgery (OR 6.64, CI 2.25–19.55), while disease-specific therapies, i.e., haematopoietic stem cell transplantation (OR 0.45, CI 0.20–1.03), produced a protective effect. Anaesthetic complications can best be predicted by surrogates for advanced disease stages and protective therapeutic factors. Further model validation in different cohorts is needed.

## 1. Introduction

Mucopolysaccharidoses (MPS), a group of inherited rare lysosomal storage disorders, are caused by reduced enzyme activity of lysosomal enzymes involved in the degradation of glycosaminoglycans (GAGs). In MPS, partially degraded GAGs accumulate in multiple organs and impair cellular function. Depending on the specific GAGs stored, MPS divide into eleven subtypes (I, II, IIIA-D, IVA and B, VI, VII and IX), each with a predominant phenotype. MPS manifests as a progressive disease and may involve craniofacial dysmorphism, cardiac and respiratory pathologies, obstructive or central sleep apnoea, hepatosplenomegaly, skeletal deformities, and neurocognitive affection [1,2]. An approved enzyme replacement therapy (ERT) exists for MPS Types I, II, IVA, VI, and VII. Hematopoietic stem cell transplantation (HSCT) is the therapy of choice in patients with MPSIH (Hurler-syndrome) below 2.5 years of age [3]. New and supplementary therapeutic approaches (e.g., gene therapy, anti-inflammatory drugs) are being tested within clinical trials.

Due to the multi-systemic morbidity, MPS patients frequently require anaesthesia for diagnostic or surgical interventions. To exemplify this, 44% of MPSI patients have undergone at least two surgeries by four years of age [4]. Simultaneously, MPS patients experience a high perioperative risk of morbidity and mortality [5,6,7]. Registry data on MPSI patients revealed a perioperative morbidity rate of up to 30% [8] and a risk of death of 0.7% within 30 days after surgery [9]. A high rate of failed direct laryngoscopy and respiratory events prove an especially difficult airway and call for advanced airway management in these patients [5,10]. We recently published that the anaesthetic risk is higher in patients with MPS types I or II than with MPSIII and suggested using indirect intubation techniques as the first-line airway approach in individuals with MPS [11].

While perioperative event rate and procedure-related risk factors have already been analysed in MPS, it has never been investigated, which disease manifestations prone for complications and to what extent. Such data are essential considering the clinical variability not only between the different MPS types but also within one type [12]. Indirect intubation techniques such as fibreoptic intubation with guidance by a supraglottic conduit are considered to be safe for the management of difficult paediatric airway [10,13], but an a priori approach may not be necessary for all MPS patients. A better prediction of the anaesthetic risk may enable a more appropriate, patient-based pre-anaesthesia strategy as well as precautionary measures and a more personalized perioperative patient management. This study aimed to develop a multivariable prediction model of anaesthesia-related complications in MPS based on available preoperative patient-specific information.

## 2. Patients and Methods

### 2.1. Study Sites and Patients

This study was performed by retrospective chart review of patients of the International Centre for Lysosomal Disorders (ICLD) located at the University Medical Centre Hamburg-Eppendorf (UKE), Hamburg, Germany. Inclusion criteria were biochemically or molecular genetically confirmed diagnosis of MPS and at least one anaesthesia procedure (procedural sedation, regional anaesthesia, general anaesthesia) performed within the framework of the clinical routine between April 2002 and October 2018, either at the UKE or the Children’s Hospital Altona (AKK), Hamburg, Germany.

### 2.2. Data Acquisition

Retrospective data acquisition was performed by systematic review of analogue and electronic patient folders (Soarian^TM^ Health Archive, Release 3.04 SP12, Siemens Healthcare, Erlangen, Germany). The latter were introduced in 2009. Of each included patient, data on the sex, the underlying subtype and the age at diagnosis were collected. The patient folders were systematically screened for information regarding the following aspects at the time of each anaesthesia (study case): body measurements, MPS-typical disease manifestations and medical history. As a surrogate parameter for the increasing experience with the disease, we also included the date and patient age at each anaesthesia.

As part of the medical history, data were collected on whether ERT (if applicable, also the age at ERT-start) or HSCT (if applicable, also the age at successful) had been performed. This study aimed to focus on anaesthesia-relevant disease manifestations, precisely, macroglossia, tonsil hyperplasia, short neck, thorax deformities (as per reports of physical examinations), spine deformities (as per reports of radiological examinations), impaired lung function (as by pulmonary function testing), sleep apnoea, respiratory infection within two weeks prior to the anaesthesia, apparent dysphagia, cardiac pathologies, relevant organomegaly (by standard deviation of the index for body length), persistent seizures, shunt-supported hydrocephalus, cognitive impairment, reduced mobility, and facial dysmorphia as an apparent symptom. Cardiac pathologies were graded as: unremarkable (incl. incomplete right bundle branch block; s/p septum defects), mild (plump valves; valvular heart defects Grades I/I–II; small septum defects; small pericardial effusion; atrioventricular block Grade I; left bundle branch block; arterial hypertension), moderate (valvular heart defects Grade II; discrete cardiac hypertrophy or dilative cardiomyopathy) and severe (valvular heart defects Grade III/IV; any manifestation requiring cardiac medication or intervention). Motor and cognitive function was graded according to the four-point scoring system (FPSS) by Meyer et al. [14]. For describing patient characteristics, the most pathologic finding counted from repeated anaesthesia procedures.

We furthermore collected anaesthesia procedure-related information. The reasons for anaesthesia were each categorised as diagnostics only, intervention, minor surgery, surgery affecting the airway, or major surgery. If anaesthesia was performed due to different reasons, the most invasive one determined classification. Punctures/injections (e.g., intravenous, catheter, lumbar), stitching, endoscopy, arthrography, change of tracheal cannulas or gastric tubes and cast installations counted as interventions. Minor surgeries comprised dental, ear and ophthalmological surgeries, catheter and gastric tube implantations, removals and revisions, herniotomies, lung and liver biopsies, abscess cleavage, circumcision, carpal tunnel syndrome releases, and surgical interventions of the lower extremities (e.g., temporary epiphysiodesis). Because of the cardiorespiratory risk, surgeries affecting the airway (e.g., tracheotomy, tonsillotomy, adenotomy) were counted as more invasive, excelled by major surgeries (i.e., thorax, cardiac, neurosurgery).

The primary endpoint “anaesthesia-related complications” was a composite of difficult airway management, respiratory events, cardiocirculatory events, and other postoperative complications and thus comprised any anaesthesia-related complication until hospital discharge. The airway management, respiratory and cardiocirculatory endpoint measures were defined as recently described [11].

### 2.3. Statistics

Data were collected in Microsoft Excel (Version 2011, Microsoft Corporation, Redmont, WA, USA) and analysed in R 4.0.3 (R core team, Vienna, Austria). Demographic data, disease manifestations, procedure-related factors and outcome data were stratified by disease type and summarised as frequencies and percentages for categorical variables and as medians and ranges or means and standard deviations (SD) for continuous variables as appropriate.

We used multivariable mixed-effects logistic regression to identify disease-specific risk factors for the composite binary endpoint “anaesthesia-related complications”. The model included one random effect to account for multiple anaesthesia procedures within one patient. Potentially eligible factors were identified by literature research, clinical implications, and descriptive and graphical analysis of the data. For variable selection least absolute shrinkage selector operator (lasso) regression was calculated [15]. The shrinkage parameter was estimated by five-fold cross-validation. For the starting model, the following variables were included for fixed effects: subtype, height, weight, age, sex, HSCT, ERT, macroglossia, tonsil hyperplasia, short neck, respiratory infection, obstructive and restrictive lung disease, sleep apnoea, cardiac manifestation, thorax deformity, hepatosplenomegaly, dysphagia, cervical spine stenosis, cervical spine immobility, spine deformity, seizures, shunt-supported hydrocephalus, cognitive and motor function impairment, reason for anaesthesia. Missing data were not imputed and cases with at least one missing value were excluded (listwise deletion). The resulting model was compared to a model, which included only the disease types (I, II, III, IV and VI) as predictors. This model was fitted by optimising the restricted maximum likelihood (REML) criterion using an iterative nonlinear optimisation algorithm as described by Bates et al., 2015 [16] and implemented in the “lme4” package for R. Odds ratios (OR) for the fixed effects are presented and respective 95% confidence intervals (CI) were calculated by Wald approximation. For categorical variables, the OR was calculated by using the category with the lowest expected event rate as reference, for example, MPSIII for the disease types. The model performance was evaluated by comparing the Akaike information criterion (AICc) and the area under the receiver operator curve (AUC). As this is an explorative study, neither model validation nor adjustment for multiple testing were performed.

## 3. Results

### 3.1. Data Acquisition and Quality

Two hundred twenty patients with MPS were screened, of which 99 underwent at least one anaesthetic procedure and were enrolled. Four cases were excluded because the anaesthesia protocol was either unreadable (*n* = 1) or unavailable (*n* = 3), adding up to 484 cases. The data quality was sufficient, with an overall missing rate of 0.75%. However, in four cases, the primary endpoint could not be defined due to insufficient data quality. Further three cases were missing primary endpoints and in 12 cases local anaesthesia was performed by a surgeon, only with anaesthesia standby. These cases had to be excluded from multivariable modelling, leaving 469 cases for the outcome analysis (Figure 1).

### 3.2. Patient Characteristics

The 99 study patients underwent a median of three anaesthetic procedures per patient (range 1–19). Sex distribution was slightly unbalanced with 71 male cases (72%). Overall, the patients were diagnosed at a median age of 3.1 years (range 0.0–29 years). The median age at first anaesthesia was 4.8 years (range 0.7–38.3 years), the median age at the last one was 9.7 years (range 0.8–38.7 years). Of the 99 patients, 35 patients had MPSI (35%), of whom 32 had MPSIH and 3 MPSIS (Scheie-syndrome). Of the 32 MPSIH patients, 22 (69%) had undergone HSCT, of whom 2 had graft failure. Another 16 patients suffered from MPSII (16%), 37 from MPSIII (37%; 27 MPSIIIA, 7 MPS IIIB, 3 MPS IIIC), 7 from MPSIV (7%; all MPS IVA) and 4 from MPSVI (4%). Overall, 36 patients received ERT (median age at start: 6.2 years, range 0.2–29 years) and 28 patients HSCT (22 MPSIH, 3 MPSII, 3 MPSIIIA; median age at HSCT: 1.3 years, range 0.4–1.8 years).

Clinical patient characteristics are described by disease type (Table 1). The most frequent symptom was facial dysmorphism (92%). Macroglossia was documented in 52% of the patients, tonsil hyperplasia in 45% and a short neck in 85%. Concerning the respiratory manifestations, 51% of the patients had impaired lung function, 39% obstructive sleep apnoea, and 26% thorax deformity. Dysphagia was apparent in 29% of the patients and 60% had at least one relevant organomegaly. Cardiac manifestations were present in 71% of the patients, of whom more than half (53%) had only mild cardiac pathologies. Overall, cervical spine pathologies were manifest in 46% of the patients and thoracolumbar spine deformity in 65%. A detailed table of cervical spine pathologies is shown in the supplement (Appendix A). Seizures persisted in 19% of the MPS patients, 5% had a shunt-supported hydrocephalus, 75% a cognitive impairment and 86% reduced mobility. Manifestations of different types overlap, especially of MPSI and II. However, the types have dominant features, particularly neurological features (seizures, cognitive impairment, immobility) in MPSIII. However, patients with MPSIII also manifest peripheral symptoms, namely, facial dysmorphism (95%), macroglossia (43%), tonsil hyperplasia (41%), short neck (78%), obstructed lung disease (32%), sleep apnoea (19%), cardiac pathologies (49%, foremost mild ones), at least one organomegaly (65%), cervical spine pathology (11%), and spine deformity (32%; foremost kyphosis and kyphoscoliosis).

### 3.3. Characteristics of Anaesthesia Procedures

Anaesthesia care was provided for 484 cases. The median age at anaesthesia was 6.1 years (IQR 3.7–11.1 years). General anaesthesia was administered in 383 cases (80%). In total, 85 patients (18%) underwent procedural sedation, with MPSIII being the type with the highest rate of procedural sedation (31%). Procedural sedation was the most common anaesthesia method for small interventions (59%). In contrast, for diagnostics (mostly magnetic resonance imaging, MRI), general anaesthesia was preferred (55.7%), while procedural sedation was less common (32%). As management was at discretion of the handling anaesthetist, the airway management was heterogeneous. Tracheal intubation was used in 69.5% of the general anaesthesia cases and a laryngeal mask airway was used in 23.6%. Tracheal intubation was facilitated by indirect methods such as videolaryngoscopy or fibreoptic intubation with or without guidance via supraglottic conduits in 71.5%. The main indications for anaesthesia were diagnostics (22%), small interventions (12%) and a broad variety of surgeries (66.5%). The anaesthesia-associated details are described by disease type in the supplement (Appendix A).

### 3.4. Anaesthesia-Related Complication

At least one anaesthesia-related complication occurred in 22.7% (*n* = 109) of the cases (Table 2). In 6.6% (*n* = 32) of the cases more than one event occurred during or after anaesthesia. The total number of events was 171 with the following event type distribution: 35.1% (*n* = 60) technically difficult airway management, 50.3% (*n* = 86) respiratory events, 7.0% (*n* = 12) cardiocirculatory events, and 7.6% (*n* = 13) other events. 45% (*n* = 77) of the events occurred during induction of anaesthesia, 17.5% (*n* = 30) during anaesthesia and 37.4% (*n* = 64) after anaesthesia. Patients with MPSIII had the lowest event rate (7.2%), whereas the highest event rate occurred in patients with MPSII (38.7%), MPSVI (31.8%), and MPSI (27.4%). Problems in airway management arose primarily due to difficult tracheal intubation (20%). Bag-mask-ventilation was described as difficult in 4.5% and the laryngeal mask placement or ventilation via laryngeal mask were problematic in 8.1% of the cases.

Until discharge, 64 postoperative events occurred in 39 study cases (22 patients) with more than one postoperative event in 41% (*n* = 16). The most frequent complications were respiratory insufficiency (*n* = 21), acute airway obstruction (*n* = 15; i.e., stridor *n* = 9, collapsed trachea *n* = 2, soft tissue swelling *n* = 1, glossoptosis *n* = 1, both-sided vocal cord paralysis *n* = 1, bronchospasm (*n* = 1), pneumonia (*n* = 6), seizures (*n* = 5), fever (*n* = 4) and atelectasis (*n* = 3). Other postoperative events comprised delirium (*n* = 2), heart failure (*n* = 2), tachycardia (*n* = 1), bronchospasm (*n* = 1), both-sided pneumothorax (*n* = 1), temporary neurological deficit (*n* = 1, MPSIH) and permanent tetraplegia (*n* = 1, MPSII). Anaesthesia-related postoperative death occurred in two cases (postoperative mortality rate: 0.4%), in both cases by cardiorespiratory decompensation on the backdrop of a difficult airway management. The first, previously described [11] patient was a 14-year-old boy with MPSII, who suffered from a severe fulminant laryngo-bronchospasm during fiberoptic intubation resulting in hypoxic cardiac arrest and death eight hours afterwards despite immediate cardiopulmonary resuscitation and emergency tracheotomy. The second patient was a 38-year-old woman with MPSVI who underwent emergency surgery of an incarcerated hernia. An extubation attempt on the first postoperative day resulted in re-intubation due to respiratory insufficiency. Tracheostomy was anatomically impossible, and weaning did not succeed. Ventilation became increasingly difficult, requiring high ventilation pressures due to a swollen and spastic airway. Based on severe aortic and mitral valve stenosis, the patient died of acute decompensated heart failure 25 days after surgery.

Within the study period of 16.7 years, the total number of anaesthesia measures increased each year, while the number of events stayed relatively stable (Figure 2). As a result, the probability of anaesthetic events decreased over time.

### 3.5. Best Forecast Performance of Anaesthesia-Related Complications by a Model Based on Disease Manifestations, Disease-Specific Therapies and the Indication for Surgery

The clinical model, which was generated by means of automatic variable selection via lasso regression (Figure 3) included 469 observations. With a shrinkage parameter of 29, the model produced the following results: HSCT (OR 0.45, CI 0.20–1.03) and ERT (OR 0.74, CI 0.36–1.55) were associated with lower risk for events. Spine deformity (OR 1.94, CI 0.80–4.71), immobility (OR 3.80, CI 0.98–14.73), obstructive lung disease (OR 1.24, CI 0.59–2.61), hepatosplenomegaly (OR 3.10, CI 1.54–6.26) and scheduled major surgery (OR 6.64, CI 2.25–19.55) were associated with an increased risk and selected for model fitting. A comparison of the effect size and the confidence intervals distinguishes hepatosplenomegaly, immobility and a planned major surgery as the most relevant predictors for anaesthesia-related complications. All other variables that had been incorporated into the starting model, including the age, gender and subtype, were dismissed as predictors during the lasso selection process. The detailed model selection path is presented in the supplement (Appendix A). The model fit and predictive power were compared to the disease subtype-model (Figure 4). Patients with MPSIII had the lowest risk for anaesthesia-related complications. Patients with MPSIH (OR 5.16, CI 2.00–13.28), MPSIS (OR 8.52, CI 1.16–62.44), MPSII (OR 9.74, CI 3.15–30.15) and MPSVI (OR 20.60, CI 3.43–123.76) suffer from a substantially increased risk compared to MPSIII patients. However, in comparison of the clinical model with the subtype-model, the addition of patient-specific information in the starting models increases the model performance (AICc 460.4 vs. 467.7; AUC 0.880 vs. 0.834).

## 4. Discussion

With 99 patients, the present study is one of the largest case series performed so far describing multi-organ manifestations in children and adults with the orphan disease MPS [17,18,19]. Moreover, with 484 anaesthetic cases in the study patients it is the largest study published as yet on anaesthetic risk in MPS and the first one to assess preoperative risk factors responsible for high peri- and postoperative morbidity [5]. In this study, almost a quarter of all cases experienced anaesthesia-related complications. According to the great phenotypical variability even within one MPS type, the best predictability power is not attributed to the underlying subtype, but to a prediction model based on the clinical manifestations and indication for anaesthesia. Precisely, this study suggests that the most relevant predictors for anaesthetic complications are hepatosplenomegaly, immobility, and planned major surgery.

By systematic chart review, this study specifies various disease manifestations in the different MPS types with a slight focus on symptoms relevant for anaesthesia planning. The natural history of MPSIII has so far only been described concerning neurocognition and behavioural patterns [20,21]. Hence, this is the first study on peripheral symptoms in MPSIII. Considering that no approved therapy exists for MPSIII, information on the natural disease course is essential to assess effects of evoking therapies. This is also one of the first studies to comprehend peripheral symptoms in a large transplanted MPSI patient cohort. Furthermore, specifics on spine disease have never been published for different disease types [22].

The prevalent MPS types vary globally. In Germany, most MPS patients suffer from MPSIII (44%), followed by MPSI (20%), MPSII (18%), MPSIV (11%) and MPSVI (7%) [12]. This order matches with the distribution pattern of this study with a shift towards MPSI as the division of paediatric stem cell transplantation of our institution is specialised in metabolic diseases. Concerning disease manifestations, it is noteworthy that airway-related symptoms such as macroglossia, sleep apnoea and airway obstruction were present throughout all MPS types, confirming the risk of difficult airway and the need for adequate precautions and postoperative monitoring potentially for all MPS patients. This is in line with a report by Berger et al. [2], who highlighted the frequency of respiratory problems in MPS, which are among the first symptoms. Patients with MPSIVA (Morquio A syndrome) can furthermore suffer from tracheal narrowing or tortuous appearance of the trachea or bronchi [23,24]. Considering the burden of spine disease in MPS [22], correct preoperative positioning is indispensable. In our study, despite precautions, two patients experienced temporary neurological deficits or even tetraplegia. Patients with MPSIVA are at particular risk for myelopathy during anaesthesia considering their skeletal phenotype with manifestations such as spinal deformities, cord compressions, adontoid hypoplasia and atlantoaxial instability, accompanied by ligamentous laxity [23,25,26]. This is supported by our data as 83% of the MPSIVA patients presented with cervical spine pathologies. Overall, 24% of the MPS patients manifested cervical spine instability and 62% cervical spine stenosis, which pose contraindications for extension of the head during intubation [5].

The overall event rate (22.7%) is consistent with the previous study performed by our centre (25.6%) [11]. The event rates vary between the different MPS types, with the highest burden in MPSII (38.7%), followed by MPSVI (31.8%) and MPSI (27.4%). For MPSIII, the event rate (7.2%) was only slightly higher than in the healthy population: 0.14–5.2% [27,28]. The overall 30-day in-hospital mortality of 0.4% is increased compared to the healthy paediatric population (0.1%) [27]. A high postoperative mortality has been described before by Arn et al., who reported on 32 of 196 deceased MPS patients, who had undergone surgery within one month of death [9].

Notably, during the study period over almost 17 years, the event rate has decreased substantially. The anaesthesia management of MPS patients hence became safer. As the ICLD is a specialised metabolic centre for MPS, a high MPS patient turnover also affects the anaesthesiologists to gain experience with this otherwise rare patient group. Actually, the necessity for an anaesthesia team experienced with MPS-specific challenges has already been deliberately emphasised on [5,25]. Nonetheless, the event rate over time should be interpreted with caution as the patient population may have changed with improved clinical management, including disease detection, specific therapies and multidisciplinary follow-up programs.

The multivariable regression models show that MPSII prones most for anaesthetic complications, followed by MPSVI and MPSI, respectively. However, additional patient-based information such as individual disease manifestations improve overall risk prediction. The model based on clinical aspects selected by the lasso regression has the best performance. In the final model, hepatosplenomegaly, immobility, and planned major surgery are the most important preoperative risk factors. Interestingly, not only the age and gender, but also the disease subtypes were dismissed as predictors in the automated parameter selection process of the clinical model. This suggests a substitution of the disease subtypes as a relevant predictor by clinical surrogate parameters. The manifestations hepatosplenomegaly and immobility imply a progressed disease stage or a poorly treated condition. Patients capable to perform lung function testing form a preselected patient group concerning age, disease type and progression (mental capacity). In case of sleep apnoea with the need for non-invasive ventilation, precautions such as postoperative transfer to the intensive care unit were taken. Therefore, respiratory manifestations might have been underestimated as predictors. As this study has an explorative design, the results of the multivariable models cannot be generalized or applied to other cohorts and hence are not designated for the clinical use. A score, by which the preoperative disease status can be translated into a predicted individual risk for anaesthetic complications requires an external validation. For this purpose, a prospective validation study is needed.

The clinical prediction model showed that disease-specific therapies might have a protective effect against anaesthesia-related complications. In the clinical multivariable model, HSCT is associated with a relevant risk reduction. Other studies strengthen our finding that HSCT decreases the overall event incidence [8,29]. Whether ERT actually has a protective effect remains unclear looking at the small effect size and large confidence interval. Other studies indicate that ERT alone does not improve difficult airway management in MPS Types I, II and VI [8,29]. Interestingly, the risk for anaesthesia-related complications was increased both in MPSIH and MPSIS to a similar extend. Thus, all individuals with MPSI are at high-risk.

In MPS patients with an expected difficult airway or cervical spine disease, intubation is typically facilitated by videolaryngoscopy or by use of a flexible bronchoscope with or without guidance via a supraglottic conduit. This approach produces the lowest conversion rate [5,11]. Our previous study suggested that the application of advanced techniques would be especially important for patients with MPS I, II, IV and VI [11,23]. Well noted, in this study, we aimed to analyse preoperative risk factors. Hence, the risk factor analysis of this study did not incorporate perioperative management factors (e.g., airway management, duration of anaesthesia). Thus, we cannot draw any concrete anaesthesia management recommendations from our findings. We previously found that airway management is the most important technical factor associated with anaesthetic events [11].

This study has few limitations. As the primary endpoint is a composite, we cannot differentiate which influencing factors exactly account for which type of adverse event. It is a retrospective study, so the documentation, also of the adverse events, may have been incomplete. Due to the high risk for anaesthetic complications in MPS, the indication for anaesthesia has to be carefully verified. For the purpose of primary endpoints, we could only include and describe patients, who actually underwent anaesthesia. The strict indication for anaesthesia and the centre experience may produce selection biases. For a better understanding and prediction of the actual risk, one would have to conduct a multi-centred, retrospective analysis of anaesthesias performed in unawareness of the risk.

## 5. Conclusions

The present study gives full particulars of the multi-morbidity of 99 MPS patients undergoing anaesthesia and is with 484 anaesthetic cases the largest cohort published so far. The rate of anaesthesia-related complications was as high as 22.7%. This study suggests that the most relevant predictors are planned major surgery, hepatosplenomegaly, and immobility, surrogate parameters for a generally advanced disease stage or an insufficiently or untreated condition. Disease-specific therapies such as HSCT and ERT produce an overall protective effect. Nonetheless, multidisciplinary preoperative diagnostics are indispensable in all patients. The burden of cardiorespiratory manifestations and spine disease call for carefully planned patient positioning and postoperative monitoring, if applicable at the intensive care unit. A better prediction of the anaesthetic risk might enable more patient-based and feasible precautionary measures. For this purpose, a prospective validation study is needed, to generate a score, by which the preoperative disease status can be translated into an individual risk assessment for anaesthetic complications.

## Figures and Tables

**Figure 1 jcm-10-03518-f001:**
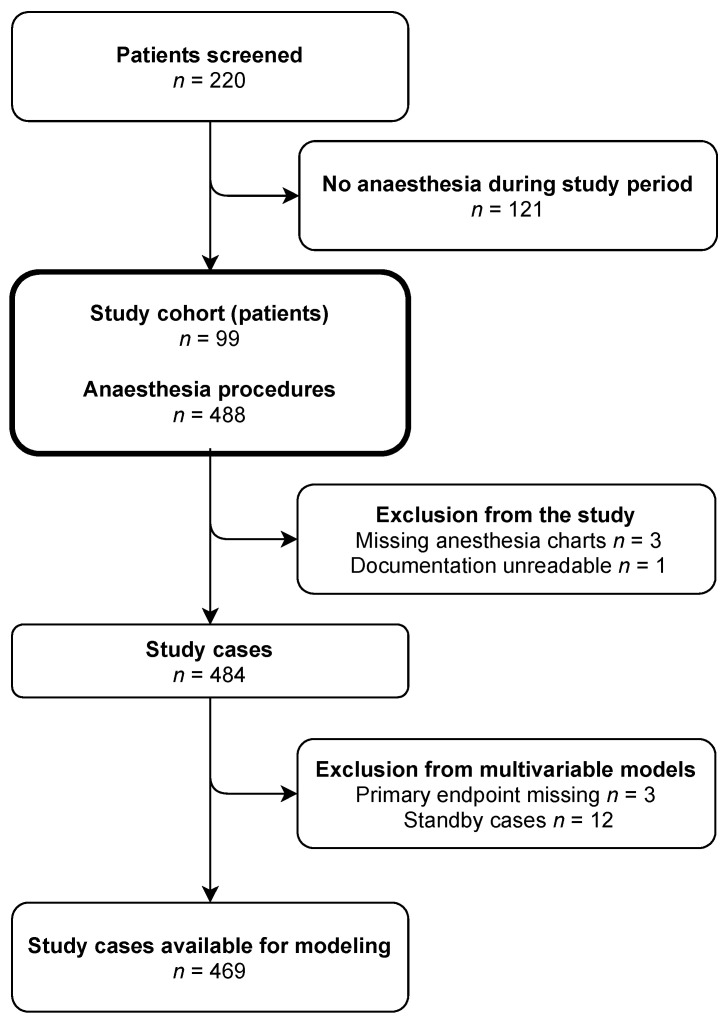
Flowchart of the study inclusion process.

**Figure 2 jcm-10-03518-f002:**
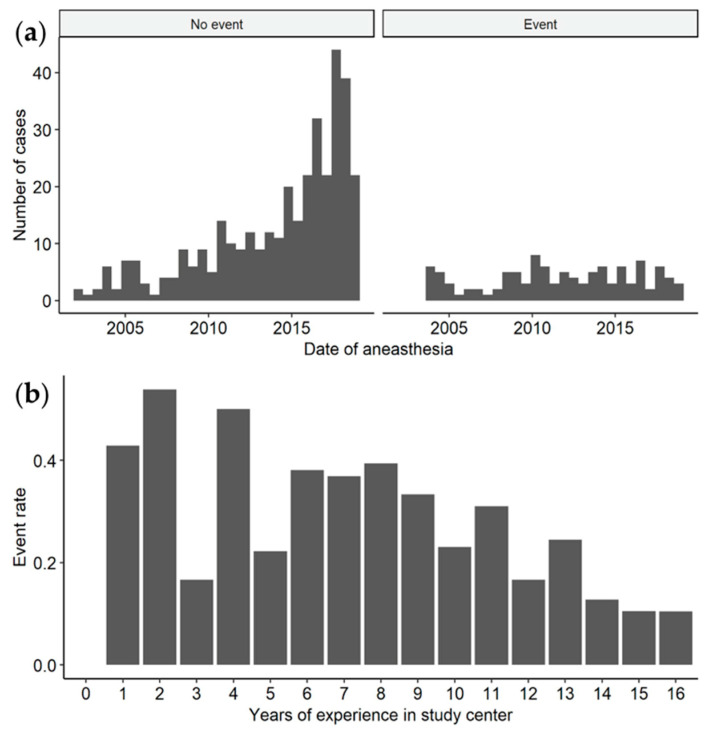
Centre experience. (**a**) Cases with and without events performed during the study period. (**b**) Event rates in different study intervals, calculated in years after inclusion of the first study patient.

**Figure 3 jcm-10-03518-f003:**
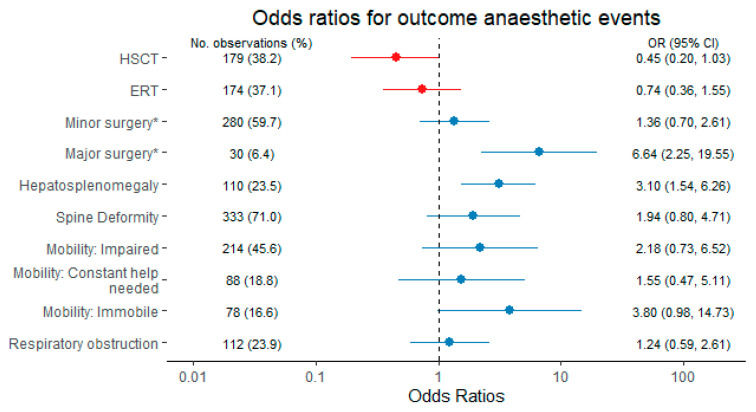
Estimated odds ratios and 95% confidence intervals (CI) for anaesthesia-related complications from a multivariable model based on automatic variable selection via lasso regression. Variables are presented in blue color (increased risk) or red color (decreased risk), respectively. * Reference: diagnostics and interventions. Abbreviations: CI, confidence interval; ERT, enzyme replacement therapy; HSCT, haematopoietic stem cell transplantation.

**Figure 4 jcm-10-03518-f004:**
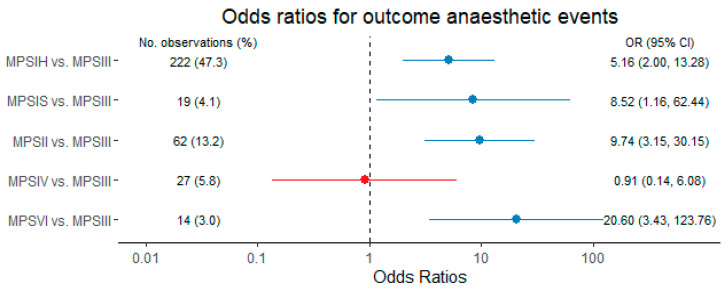
Estimated odds ratios and 95% confidence intervals (CI) for anaesthesia-related complications from a multivariable model based on disease subtypes. Variables are presented in blue color (increased risk) or red color (decreased risk), respectively. Abbreviations: CI, confidence interval; MPS, mucopolysaccharidosis.

**Table 1 jcm-10-03518-t001:** Patient characteristics stratified by MPS disease types.

Characteristics	Overall *N* = 99	MPSIH *N* = 32	MPSIS *N* = 3	MPSII *N* = 16	MPSIII *N* = 37	MPSIV *N* = 7	MPSVI *N* = 4
Demographic characteristics							
Sex, male, *n* (%)	71 (71.7)	20 (62.5)	1 (33.3)	16 (100.0)	27 (73.10)	4 (57.1)	3 (75.0)
Min. age at anaesthesia (years) *	4.8 (0.7–38.3)	1.9 (0.8–20.8)	13.9 (7.1–29.1)	7.8 (0.7–29.7)	5.8 (1.1–38.3)	8.6 (4.1–18.6)	10.4 (7.5–24.2)
Max. age at anaesthesia (years) *	9.7 (0.8–38.7)	8.6 (0.8–23.8)	24.0 (7.1–29.1)	12.2 (2.5–32.1)	8.9 (3.5–38.7)	15.7 (6.1–18.6)	21.6 (8.5–38.2)
Medical history							
Age at diagnosis (years) *	3.1 (0.0–29.0)	1.2 (0.0–8.0)	6.6 (3.8–29.0)	4.5 (0.1–15.8)	3.7 (0.5–16.8)	4.3 (0.9–8.6)	4.4 (1.6–11.5)
Total no. of anaesthesias	3.0 (1.0–19.0)	5.0 (1.0–19.0)	1.0 (1.0–18.0)	3.0 (1.0–10.0)	2.5 (1.0–12.0)	3.0 (1.0–9.0)	5.5 (1.0–11.0)
HSCT, *n* (%)	28 (28.3)	22 (68.8)	-	3 (18.8)	3 (8.1)	-	-
Age at HSCT (years) *	1.3 (0.4–1.8)	1.4 (1.1–1.8)	N/A	0.4 (0.4–0.4)	4.7 (3.0–5.7)	N/A	N/A
ERT, *n* (%)	36 (36.4)	15 (46.9)	3 (100.0)	11 (68.8)	-	5 (71.4)	3 (75.0)
Age at ERT-start (years) *	6.2 (0.2–29.0)	1.0 (0.7–6.4)	7.0 (6.7–29.0)	8.2 (0.2–22.1)	N/A	12.0 (4.6–14.5)	11.5 (11.5–11.5)
Craniofacial pathologies							
Facial dysmorphism, *n* (%)	91 (91.9)	30 (93.8)	2 (66.7)	15 (93.8)	35 (94.6)	5 (71.4)	4 (100.0)
Macroglossia, *n* (%)	51 (51.5)	20 (62.5)	1 (33.3)	11 (68.8)	16 (43.2)	1 (14.3)	2 (50.0)
Tonsil hyperplasia, *n* (%)	45 (45.5)	17 (53.1)	-	6 (37.5)	15 (40.5)	4 (57.1)	3 (75.0)
Short neck, *n* (%)	84 (84.8)	27 (84.4)	2 (66.7)	16 (100.0)	29 (78.4)	6 (85.7)	4 (100.0)
Respiratory manifestations							
Lung function, *n* (%)							
Normal	49 (49.5)	13 (40.6)	2 (66.7)	4 (25.0)	25 (67.6)	4 (57.1)	1 (25.0)
Obstruction	23 (23.2)	6 (18.8)	-	4 (25.0)	12 (32.4)	1 (14.3)	-
Restriction	15 (15.2)	12 (37.5)	-	1 (6.2)	-	1 (14.3)	1 (25.0)
Both	12 (12.1)	1 (3.1)	1 (33.3)	7 (43.8)	-	1 (14.3)	2 (50.0)
Sleep apnoea, *n* (%)							
No	60 (60.6)	18 (56.2)	1 (33.3)	7 (43.8)	30 (81.1)	4 (57.1)	-
Suspected	17 (17.2)	5 (15.6)	-	4 (25.0)	5 (13.5)	1 (14.3)	2 (50.0)
Diagnosis	11 (11.1)	5 (15.6)	1 (33.3)	3 (18.8)	1 (2.7)	-	1 (25.0)
CPAP	11 (11.1)	4 (12.5)	1 (33.3)	2 (12.5)	1 (2.7)	2 (28.6)	1 (25.0)
Thorax deformity, *n* (%)	26 (26.3)	11 (34.4)	-	4 (25.0)	3 (8.1)	6 (85.7)	2 (50.0)
Cardiac manifestations							
Cardiac pathology, *n* (%)							
No	29 (29.3)	5 (15.6)	-	3 (18.8)	19 (51.4)	2 (28.6)	-
Mild	38 (38.4)	13 (40.6)	2 (66.7)	5 (31.2)	12 (32.4)	4 (57.1)	2 (50.0)
Moderate	15 (15.2)	4 (12.5)	1 (33.3)	5 (31.2)	3 (8.1)	1 (14.3)	1 (25.0)
Severe	17 (17.2)	10 (31.2)	-	3 (18.8)	3 (8.1)	-	1 (25.0)
Gastrointestinal manifestations							
Organomegaly, *n* (%)							
No	40 (40.4)	11 (34.4)	2 (66.7)	5 (31.2)	13 (35.1)	7 (100.0)	2 (50.0)
Any	23 (23.2)	8 (25.0)	-	1 (6.2)	13 (35.1)	-	1 (25.0)
Hepatosplenomegaly	36 (36.4)	13 (40.6)	1 (33.3)	10 (62.5)	11 (29.7)	-	1 (25.0)
Dysphagia, *n* (%)	29 (29.3)	4 (12.5)	1 (33.3)	5 (31.2)	19 (51.4)	-	-
Spine disease							
Cervical spine stability, *n* (%)							
Stable	75 (75.8)	18 (56.2)	3 (100.0)	13 (81.2)	37 (100)	3 (42.9)	1 (25.0)
Instable	19 (19.2)	11 (34.4)	-	3 (18.8)	-	3 (42.9)	2 (50.0)
Surgical Fusion	5 (5.1)	3 (9.4)	-	-	-	1 (14.3)	1 (25.0)
Cervical spine stenosis, *n* (%)							
No	57 (57.6)	11 (34.4)	1 (33.3)	11 (68.8)	33 (89.2)	1 (14.3)	-
Stenosis	25 (25.3)	12 (37.5)	-	3 (18.8)	4 (10.8)	4 (57.1)	2 (50.0)
Stenosis with myelopathy	6 (6.1)	3 (9.4)	1 (33.3)	2 (12.5)	-	-	-
Decompression surgery	11 (11.1)	6 (18.8)	1 (33.3)	-	-	2 (28.6)	2 (50.0)
Spine deformity, *n* (%)							
No	35 (35.4)	1 (3.1)	1 (33.3)	8 (50.0)	25 (67.6)	-	-
Scoliosis	10 (10.1)	2 (6.2)	-	5 (31.2)	2 (5.4)	-	1 (25.0)
Kyphosis	26 (26.3)	13 (40.6)	-	1 (6.2)	7 (18.9)	3 (42.9)	2 (50.0)
Both	28 (28.3)	16 (50.0)	2 (66.7)	2 (12.5)	3 (8.1)	4 (57.1)	1 (25.0)
Neurological manifestations							
Seizures, *n* (%)	19 (19.2)	3 (9.4)	-	2 (12.5)	14 (37.8)	-	-
Shunted hydrocephalus, *n* (%)	5 (5.1)	3 (9.4)	1 (33.3)	-	-	-	1 (25.0)
Cognitive function, *n* (%)							
Normal	25 (25.3)	11 (34.4)	2 (66.7)	5 (31.2)	-	5 (71.4)	2 (50.0)
Impaired	46 (46.5)	18 (56.2)	1 (33.3)	7 (43.8)	16 (34.2)	2 (28.6)	2 (50.0)
Regression	15 (15.2)	2 (6.2)	-	4 (25.0)	9 (24.3)	-	-
Unresponsiveness	13 (13.1)	1 (3.1)	-	-	12 (32.4)	-	-
Motor function, *n* (%)							
No impairment	14 (14.1)	1 (3.1)	-	5 (31.2)	8 (21.6)	-	-
Impaired	34 (34.3)	16 (50.0)	1 (33.3)	6 (37.5)	7 (18.9)	2 (28.6)	2 (50.0)
Constant help needed	23 (23.2)	7 (21.9)	1 (33.3)	1 (6.2)	11 (29.7)	3 (42.9)	-
Immobile	28 (28.3)	8 (25.0)	1 (33.3)	4 (25.0)	11 (29.7)	2 (28.6)	2 (50.0)

* Median (range). Abbreviations: CPAP, continuous positive airway pressure device; ERT, enzyme replacement therapy; HSCT, haematopoietic stem cell transplantation.

**Table 2 jcm-10-03518-t002:** Anaesthesia-related complications stratified by MPS disease types.

Characteristics	Overall *N* = 484	MPSIH *N* = 224	MPSIS *N* = 20	MPSII *N* = 62	MPSIII *N* = 126	MPSIV *N* = 29	MPSVI *N* = 23
Respiratory events, *n* (%)	86 (17.8)	38 (17.0)	6 (30.0)	25 (40.3)	7 (5.6)	2 (6.9)	8 (34.8)
Respiratory insufficiency	23 (4.8)	8 (3.6)	2 (10.0)	7 (11.3)	2 (1.6)	1 (3.4)	3 (13.0)
Hypoxemia	17 (3.5)	11 (4.9)	1 (5.0)	2 (3.2)	2 (1.6)	-	1 (5.0)
Airway obstruction	19 (3.9)	6 (2.7)	2 (10.0)	8 (12.9)	1 (0.8)	-	2 (10.0)
Increased ventilation pressure	8 (1.7)	2 (0.9)	-	4 (6.5)	-	1 (3.4)	1 (5.0)
Hypercapnia	7 (1.4)	7 (3.1)	-	-	-	-	-
Pneumonia	6 (1.2)	2 (0.9)	1 (5.0)	2 (3.2)	-	-	1 (5.0)
Atelectasis	4 (0.8)	2 (0.9)	-	-	2 (1.6)	-	-
Pneumothorax	1 (0.2)	-	-	1 (1.6)	-	-	-
Cardiocirculatory events, *n* (%)	12 (2.5)	3 (1.3)	-	1 (1.6)	4 (3.2)	-	4 (17.4)
Bradycardia/tachycardia	6 (1.2)	2 (0.9)	-	1 (1.6)	3 (2.4)	-	-
Hypotension	3 (0.6)	1 (0.4)	-	-	-	-	2 (8.7)
Heart failure	3 (0.6)	-	-	-	1 (0.8)	-	2 (8.7)
Difficult airway management, *n* (%)	60 (12.4)	37 (16.5)	5 (25.0)	13 (21.0)	2 (1.6)	-	3 (13.0)
Technique changes necessary	31 (6.4)	22 (9.8)	1 (5)	5 (8.1)	1 (0.8)	-	2 (8.7)
Blind intubation	12 (2.5)	7 (3.1)	1 (5)	3 (4.8)	1 (0.8)	-	-
Primary technique difficulty	11 (2.3)	7 (3.1)	2 (10)	2 (8.1)	-	-	-
Airway could not be secured	4 (0.8)	-	1 (5)	2 (8.1)	-	-	1 (4.3)
Prolonged ventilation due to difficult airway	2 (0.4)	1 (0.4)	-	1 (1.6)	-	-	-
Other Events, *n* (%)	13 (2.7)	5 (2.2)	1 (5.0)	4 (6.5)	2 (1.6)	-	1 (4.3)
Seizures	5 (1.0)	1 (0.4)	-	2 (3.2)	2 (1.6)	-	-
Fever	4 (0.8)	3 (1.3)	-	-	-	-	1 (4.3)
Delirium	2 (0.4)	-	1 (5.0)	1 (1.6)	-	-	-
Neurological residues	2 (0.4)	1 (0.4)	-	1 (1.6)	-	-	-

## Data Availability

The data that support the findings of this study are available from the corresponding author, upon reasonable request. The data are not publicly available to ensure, that the privacy of the study patients with rare diseases is not compromised.

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
