# Peer review of "Disease Manifestations in Mucopolysaccharidoses and Their Impact on Anaesthesia-Related Complications—A Retrospective Analysis of 99 Patients"

_jcm, 2021, doi:10.3390/jcm10163518_

Round 1

Reviewer 1 Report

The manuscript deals with the anaesthesic risk that is one of the big issues during MPS management. The question of the predicted risk is essential

Introduction: Line 68: "save" or safe?

Methods:

The study design excludes patients with an high risk for anaesthesia a priori. This is a major bias that could have been discussed later. Also, it could have been interesting to mention all the different reasons that could have prevent anaesthesia. It appears necessary to discuss this point whereas authors evoke an increasing number of anaesthesia : are the patients still the same.

Results:

It is difficult to figure out what mean the « Anaesthetic adverse events» in terms of severity (the post operative events are much more detailed and explicit). I have to admit that I'm quite disappointed not to find in the results the different factors predicitve for "the problem in tracheal intubation" line 233.

Conlusion

The findings are quite disappointing: the more the patients are severe, the more risky is anaesthesia.

The study seems not to answer the question it raises. Patients are mixed in terms of diseases, of age, of symptoms but also in terms of reason for anaesthesia. It is a source of big mix-up. We are all afraid about in/extubation problems and spinal problems associated with anaesthesia in MPS patients which are not clearly addressed here in terms of prediction. It is essential to add some data

The design of the study leads to numerous bias

Reviewer 2 Report

Patients with MPS are at high risk for anesthesia-related complications and require a high level of attention. The paper authored by Ammer et al. is a kind of statistical categorization of patients with various MPS types and anesthesia-related complications. The discussion is lack of explanations (from the clinical point of view) of Authors' results. 

Statistical analyses are not needed to know that patients with MPSIII had the  lowest event rate of anesthesia-related complications. MPS I, II, and VI show very similar aspects; MPS IVA also shares these characteristics, although skeletal involvement is more apparent. Patients with MPS III may also have disease processes which can complicate anesthesiological management, although somatic manifestations are generally less severe, and therefore the lowest rate of anesthesia-related complications is observed.

In patients with MPS IV the skeletal system is primarily affected. Multiple abnormalities subject the patient to higher (than other MPS types) anesthetic risk. It is also a well known fact. Why did the Authors not cite the relevant studies??

It is also well known that ERT/HSCT are associated with lower incidence of airway complications/difficult airway management related to general anesthesia.

The paper is well-written and important for MPS community, thus I propose the Authors to revise their manuscript.

Author Response

Response to Reviewer 2 Comments
Patients with MPS are at high risk for anesthesia-related complications and require a high level of attention. The paper authored by Ammer et al. is a kind of statistical categorization of patients with various MPS types and anes-thesia-related complications. The discussion is lack of explanations (from the clinical point of view) of Authors' results.
1. Statistical analyses are not needed to know that patients with MPSIII had the lowest event rate of anes-thesia-related complications. MPS I, II, and VI show very similar aspects; MPS IVA also shares these char-acteristics, although skeletal involvement is more apparent. Patients with MPS III may also have disease processes which can complicate anesthesiological management, although somatic manifestations are gen-erally less severe, and therefore the lowest rate of anesthesia-related complications is observed.
Thank you for the close reading of our manuscript and for your valuable input. We agree that it had already been well-known that patients with MPSIII experience the lowest rate of perioperative complications (Dohrmann et al. 2020). Moreover, it is even comparable to the risk of healthy children (Habre et al. 2017, Kurth et al. 2014). That is why we chose MPSIII as our reference point for risk calculations. Either way, we wanted to confirm the relatively low risk of MPSIII patients in our own patient group, which is one of the largest case series on anaesthetic risk in MPS so far.
2. In patients with MPS IV the skeletal system is primarily affected. Multiple abnormalities subject the patient to higher (than other MPS types) anesthetic risk. It is also a well known fact. Why did the Authors not cite the relevant studies??
Thank you for pointing out background information and references that had been missing. In the discussion, we added:
“Patients with MPSIVA (Morquio A syndrome) can furthermore suffer from tracheal narrowing or tortous appearace of the trachea or bronchi (Theroux et al. 2012, Averill et al. 2021). Considering the burden of spine disease in MPS (Remondino et al. 2019), correct preoperative positioning is indispensable. In our study, despite precautions, two patients experienced temporary neurological deficits or even tetraplegia. Patients with MPSIVA are at particular risk for myelopathy during anaesthesia considering their skeletal phenotype with manifestations such as spinal deformities, cord compressions, adontoid hypoplasia and atlantoaxial instability, accompanied by ligamentous laxity (23, 25, 26). This is supported by our data as 83% of the MPSIVA patients presented with cervical spine pathologies.”
3. It is also well known that ERT/HSCT are associated with lower incidence of airway complications/difficult airway management related to general anesthesia. The paper is well-written and important for MPS com-munity, thus I propose the Authors to revise their manuscript.
Thank you for your interest in this subtopic. We agree with you; the effect of causal therapies on the an-aesthetic risk has already been well investigated (Frawley et al. 2012, Kirkpatrick et al. 2012, Megens et al. 2015). Especially HSCT decreases intubation and airway difficulties (Kirkpatrick et al. 2012, Frawley et al. 2012). Our literature research revealed that the effect of ERT on perioperative complications is less clear. Kirkpatrick et al. reported on 39 patients with MPSIH. The more attenuated patients receiving ERT were still at high risk for airway (57%) and intubation (3%) problems. Frawley at al. suggested that ERT ameliorates intubation, but not the airway management of patients with MPSII and VI.
We would furthermore like to point out that the endpoints of the studies cited differ from ours. Besides intubation and respiratory complications, we also included cardiological and postoperative anaesthesia-related complications. The results of our study are in line with the literature. Nonetheless, considering the different endpoints, comparisons of our study to the literature should be made with caution.

Round 2

Reviewer 1 Report

I really appreciate the table 2 that points out the different complications observed during anaesthesia stratified by MPS subgroup. It is useful to discuss anaesthesia with patients, families and colleagues. 

I understand the inherent limitations associated with rare diseases...

Reviewer 2 Report

I appreciate the hard work made by Authors to collect all the data, write a manuscript, and also answer to my comments.

The great value of this manuscript is that it is very important to MPS community. The data presented are only a subtle variation on previous work with the same overall scientific message. However, the strenght is the large cohort of patients, inluding various MPS types.

Did the Authors find differences between children and adults with MPS regarding anaesthesia-related complications. This issue need to be answered before publication. Aneesthesia-related complications stratified by MPS disease types and the age of patients will be essential.

Author Response

I appreciate the hard work made by Authors to collect all the data, write a manuscript, and also answer to my comments. The great value of this manuscript is that it is very important to MPS community. The data presented are only a subtle variation on previous work with the same overall scientific message. However, the strenght is the large co-hort of patients, inluding various MPS types. Did the Authors find differences between children and adults with MPS regarding anaesthesia-related complica-tions. This issue need to be answered before publication. Aneesthesia-related complications stratified by MPS disease types and the age of patients will be essential.
Thank you for the kind summary. We are greateful for the quick and positive response.
We would also be interested in being able to stratify anaesthesia-related complications by age. Unfortunately, such analysis is challenging with the limited case numbers in the rare disease MPS. Descriptive statistics and simple tests of significance are not possible as there are multiple anaesthesias per patient. This especially applies to subgroups with low patient numbers. We included only a few adult patients (n = 18), some with more than ten anaesthesia procedures and multiple anaesthesia-related complications. Taken together, the inclusion of a heter-ogenous population in this cross-sectional study in terms of demographics and medical history as well as repeated measures can produce biases and need special attention.
In our study, the impact of each variable on the outcome was assessed in a multivariable approach. Each variable was confirmed or dismissed as a predictor for anaesthesia-related complications by automated variable selection (lasso method). We hence think that we might have managed to rule out confounders for our real-world data. When looking at the event rate descriptions, there might be a tendency for more events in the adult population. However, we want to be careful not to overinterpret the results of differentiated subanalyses. In the lasso regression model, among others, age was dismissed as a relevant influencing factor during the selection process (Figure S1). So, after all, we did not detect any relevant difference between children and adults concerning the risk for anaesthesia-related complications when correcting for multiple confounders and the repeated events within single patients.
To clarify this, we added the following in the results section: “All other variables that had been incorporated into the starting model, including the age, gender and subtype, were dismissed as predictors during the lasso selec-tion process.”
In the discussion, we updated: “Interestingly, not only the age and gender, but also the disease subtypes were dismissed as predictors in the automated parameter selection process of the clinical model. This suggests a sub-stitution of the disease subtypes as a relevant predictor by clinical surrogate parameters.”
